# The Histopathological “Placentitis Triad” Is Specific for SARS-CoV-2 Infection, and Its Acute Presentation Can Be Associated with Poor Fetal Outcome

**DOI:** 10.3390/life13020479

**Published:** 2023-02-09

**Authors:** Annabelle Remoué, Yurina Suazo, Marie Uguen, Arnaud Uguen, Pascale Marcorelles, Claire de Moreuil

**Affiliations:** 1Service d’Anatomie et Cytologie Pathologiques, CHRU Morvan, 2 Avenue Foch, 29609 Brest, France; 2CHRU de Brest, LBAI (Lymphocytes B, Autoimmunité et Immunotherapies), University of Brest, Inserm, UMR 1227, 29200 Brest, France; 3CHU de Brest, LIEN (Laboratoire Interactions Epithéliums Neurones), University of Brest, EA 4586, 29200 Brest, France; 4Département de Médecine Vasculaire, Médecine Interne et Pneumologie, CHRU de Brest, 29609 Brest, France; 5CHRU de Brest, GETBO (Groupe d’Etude de la Thrombose de Bretagne Occidentale), University of Brest, Inserm, UMR 1304, 29200 Brest, France

**Keywords:** placenta, COVID-19, SARS-CoV-2, massive perivillous fibrin deposition, intervillositis, prematurity

## Abstract

(1) Background: Placental histological lesions reported in relation with SARS-CoV-2 infection are various, with potential consequences such as fetal growth retardation, prematurity or stillbirth/neonatal death. We report here on a placental pathological association which could be specific for SARS-CoV-2 infection and associated with poor fetal outcome; (2) Methods: We collected all the placental pathological examinations performed in Brest University Hospital (France) since the beginning of COVID-19 pandemic with a known maternal SARS-CoV-2 infection and a poor pregnancy outcome. In these cases, we described the pathological lesions and we searched for these lesions in a large series of placentas collected and examined in the same institution before the SARS-CoV-2 pandemic; (3) Results: Three cases with severe fetal outcome (tardive abortion, prematurity, neonatal death), from the first to the third trimesters of pregnancy, were included. The three cases showed features of massive and acute “placentitis triad” consisting in massive perivillous fibrin deposition, sub-acute intervillositis and trophoblastic necrosis. This association was not encountered in any of 8857 placentas analyzed during the period between 2002 and 2012 in our institution; (4) Conclusions: The “placentitis triad” appears to be specific for SARS-CoV-2 infection and, in case of massive and acute presentation, could result in poor fetal outcome.

## 1. Introduction

Since the emergence of SARS-CoV-2 virus, responsible for COVID-19 disease, the effects of SARS-CoV-2 infection on the outcome of pregnancy have been analyzed in several publications, with different results. A meta-analysis by Khalil et al. dedicated to SARS-CoV-2 infected pregnant women has notably concluded in rates of 21.8% prematurity and 0.6% neonatal death, and a literature review by Juan et al. has reported 1.4% miscarriage/abortion and 0.5% neonatal death [1,2]. Some studies have also described the diminution of intra-uterine fetal movements in the context of acute maternal SARS-CoV-2 infection, resulting, in one study, in two cases of severe brain injuries, and, in two other studies, in two cases of fetal deaths [3,4,5,6]. This highlights the potential dramatic consequences of SARS-CoV-2 virus infection on fetal outcome and points to the need for a better understanding of its pathophysiological bases. Our work was conducted for this purpose.

Some pathophysiological hypotheses have already been generated to explain SARS-CoV-2 effects on pregnancy outcomes through transplacental fetal infection and changes in placental homeostasis. For example, SARS-CoV-2 infection is linked to Angiotensin Converting Enzyme 2 (ACE2) and Transmembrane Protease Serine 2 precursor (TMPRSS2), whose expression decreases with gestational age, which could explain a higher transplacental transmission of the virus during the first trimester of pregnancy than during the second and third trimesters [7]. Supporting this hypothesis, among the placentas of 20 pregnant women (all at second and third trimesters of pregnancy) with positive SARS-CoV-2 PCR nasopharyngeal tests, Celik and al. have detected no SARS-CoV-2 PCR positive placenta [8]. In addition to the term of pregnancy, some authors have also postulated that the placental involvement is only transient during SARS-CoV-2 maternal infection, and that the detection of the virus in the placenta tissue could decrease in case of long delay between the contamination of the placenta tissue, the delivery and placental analysis [3,9]. Despite its well-demonstrated transplacental transmissibility, no teratogenous effect of SARS-CoV-2 infection has been reported to date and the effect of SARS-CoV-2 infection on pregnancy outcome seems to be correlated with the existence of inflammatory placental lesions in most studies.

From a pathological point of view, placental lesions among pregnant women with SARS-CoV-2 infection are reported to range from a placenta devoid of any macroscopic or microscopic change to various non-specific lesions [10,11]. These non-specific lesions comprise, on the one hand, lesions related to maternal vascular malperfusion, such as accelerated villous maturation, villous infarction and intervillous thromboses, and on the other hand, chronic or subacute inflammatory lesions such as chorioamniotis, villitis, intervillositis or massive perivillous fibrin deposition (MPFD). Fetal vasculopathy is also sometimes reported in the literature [12,13,14]. The frequencies of these different lesions were summarized in some meta-analyses. First, Sharp et al. analyzed the data of 50 studies on 328 placentas in the context of SARS-CoV-2 infection, independently of the detection of SARS-CoV-2 in the placenta, and reported 46% maternal vascular malperfusion, 35.3% fetal vascular malperfusion, 8.7% villitis, 5.3% intervillositis, 6% acute chorioamniotis and 30% fibrin deposition [15]. A second meta-analysis, by Wong et al., focused only on studies with evidence of SARS-CoV-2 virus detected in the placenta and concluded in 37.8% maternal vascular malperfusion, 9.2% fetal vascular malperfusion, 43.2% increased fibrin deposition, 29.4% intervillositis, 14.7% villitis, 35.3% acute chorioamniotis, 35.1% infarction and 5.4% thrombosis [12]. Finally, in a third meta-analysis, Girolamo et al. analyzed the pathological data of 1009 cases of SARS-CoV-2 infected women from 57 studies and concluded in 31.4% maternal vascular malperfusion lesions, 26.9% fetal vasculopathy, 32.7% MPFD and 14.5% intervillous thromboses, whereas acute and chronic inflammation were observed in 22.6% and 26.2% of cases, respectively [16].

Beyond the individual lesions themselves, some authors have also proposed a placental histological SARS-CoV-2 signature associating different lesions, also called “placentitis triad”, with the association of (1) chronic histiocytic intervillositis, (2) excess fibrin deposition and (3) trophoblast necrosis [4,9,17,18,19,20,21]. Focusing on placentas of SARS-CoV-2 infected women with poor fetal outcome, among five cases (with three cases of fetal deaths and two cases with severe prematurity at 26 weeks of gestation (WG), four out of five cases with less than 10 days between SARS-CoV-2 infection and fetal complications), Bouachba et al. have also reported the association of MPFD, chronic histiocytic intervillositis in addition to intervillous thromboses. [10]. We found that chronic histiocytic intervillositis is a rare histological lesion, with an estimated prevalence of 6 out of 10,000 placentas (0.03–0.5%), with diagnostic criteria established by Bos in 2017 [22,23,24].

Investigating the specificity of the “placentitis triad” in relation with SARS-CoV-2 infection was the aim of the present study. Indeed, beyond reporting on three new cases with severe fetal outcome in SARS-CoV-2 infected women, with this particular microscopic pattern of placental lesions, we also searched in a second time for this “placentitis triad” among placentas analyzed previously in our institution, at a time when SARS-CoV-2 virus did not exist, before the COVID-19 pandemic. Albeit, this particular association would not be sensitive for the diagnosis of SARS-CoV-2-related placental lesions, ranging from almost normal to various pathological features, the “placentitis triad” could be specific enough to suggest the role of SARS-CoV-2 infection in the poor outcome of some pregnancies.

## 2. Materials and Methods

### 2.1. First Step: Inclusion of SARS-CoV-2 Infected Cases

From the placentas referred for pathological examination to the department of Pathology of Brest University Hospital since the beginning of the COVID-19 pandemic, between January 2020 and August 2021, we collected the cases associated (1) with a context of poor fetal or neonatal outcome and (2) arising in women with confirmed SARS-CoV-2 infection (positive nasopharyngeal PCR test), and selected three particular cases, whose characteristics are summarized in Table 1. In the case of the 16 WG abortion, the fetal autopsy was performed after collecting the parent’s consent. Macroscopic and microscopic placental examination were performed following standardized protocol and written report used in placental pathology for more than 20 years in the department of Pathology of Brest University Hospital. In addition to the initial Hematoxylin Eosin Saffron (HES) used for histopathological analyses of four placental formalin-fixed and paraffin-embedded samples, immunochemistry analyses were performed on samples with HES-identified inflammatory infiltrate using anti-CD3 antibody (polyclonal, 1:100 dilution, Dako, Santa Clara, CA, USA), anti-CD68 antibody (clone PG-M1, 1:100 dilution, Dako), anti-myeloperoxydase antibody (polyclonal, 1:300, Dako), anti-CD20 antibody (clone L26, 1:100 dilution, Dako) and anti-SARS-CoV-2 N protein antibody (clone AMC0368, 1:100 dilution, ABclonal, Woburn, MA, USA). Immunohistochemistry analyses were performed on Ventana Benchmark ultra automaton (Roche Diagnostics, Meylan, France). For the anti-CD3, anti-CD20, anti-CD68 and anti-myéloperoxydase antibodies, we looked for a brown positivity, interpreted as a positive staining in favor of presence of T lymphocytes, B lymphocytes, macrophages or neutrophils, respectively. The quantitative evaluation of each antibody was subjective, based on the pathologist experience, on the abundance of positive stain cells, and classified into three categories: «low» for a minor positive staining, «moderate» for a moderate positive staining and «high» for an abundant positive staining. For the anti-SARS-CoV-2 antibody, we looked for a positive stain on the trophoblastic cytoplasm of the samples concerned by the placentitis triad, in the areas where intervillositis was described with HES coloration.

### 2.2. Second Step: Comparison with Placentas Analyzed before the Emergence of SARS-CoV-2 Virus (Historical Cohort)

In order to search for the “placentitis triad” association out of SARS-CoV-2 infected cases, we led a retrospective study based on the analysis of standardized written reports of placental examination in the same institution, searching for this association of lesions (intervillositis, fibrin deposition and trophoblast necrosis) at a time when SARS-CoV-2 and COVID-19 did not exist (from January 2002 to June 2012, case series of the department of Pathology of Brest University Hospital), with a special focus on cases reporting intervillositis among other placental inflammatory lesions. All samples were included in a registered tissue collection and the present study was conducted after approval by our institutional review board (CHRU Brest, CPP n° DC-2008-214).

### 2.3. Statistics

Cumulative incidence rates of “placentitis triad” and 95% confidence intervals (95% CIs) during the two studied periods (before COVID-19 pandemic and during COVID-19 pandemic) were calculated under the Poisson distribution assumption.

## 3. Results

### 3.1. SARS-CoV-2 Infected Cases

Three placentas analyzed in the context of maternal SARS-CoV-2 infection and severe fetal outcome were included in the study, occurring from the first to the third trimesters of pregnancy, and associated with tardive abortion (one case) or prematurity (two cases, including subsequent neonatal death in one case), over 1852 placentas examined during the study period. The cumulative incidence rate of the “placentitis triad” during this period was then 0.16% (95% CI 0.05–0.47%). The most frequent SARS-CoV-2 variant in France, when these observations were collected, was the Delta variant, which is the SARS-CoV-2 variant biologically confirmed in the second case. The data for the three cases are summarized in Table 1 and illustrated in Figure 1, Figure 2 and Figure 3. Pregnancy follow-up was normal, without obstetrical background for the two first observations. In the third case, the mother presented a maternal thrombopenia, leading to an emergency platelet transfusion before the delivery by cesarean section.

In all cases, the placenta had no hypotrophy, with a normal weight for gestational age. In all cases, macroscopic examination revealed massive and diffuse intraparenchymal dense and greyish fibrin deposition, with rare no pathological areas, confirmed at microscopic examination (Figure 1, Figure 2 and Figure 3). Trophoblast necrosis and intervillositis were constant in all cases, with not only mononuclear histiocytic and lymphocytic infiltrates, but also constant neutrophils infiltrate, with variable proportions of these inflammatory cells populations. Traditionally, in chronic histiocytic intervillositis, there is no neutrophils described. In addition to MPO+ neutrophils, CD68+ macrophages and CD3+ T cells, CD20+ B cells were also present in the intervillous space in two cases (Figure 1F and Figure 3F). In all cases, these inflammatory lesions were observed in at least 50% of the placental parenchyma. No cytopathogenic effect was observed histologically. There was no clinical or histological argument in favor of another infectious disease, such as toxoplasmosis, rubella, cytomegalovirus infection, herpes simplex virus, syphilis or HIV. For all cases, we performed a complementary analysis with the anti-SARS-CoV-2 antibody: all cases showed a syncytiotrophoblast cytoplasmic positivity, only in the placental areas showing features of placentitis triad (including intervillositis which can be observed), as shown in Figure 4. The areas of the three cases not concerned by intervillositis and placentitis triad showed no positive staining.

The autopsic examination of case 1 at 16 WG revealed no malformation and menstruations were normal for gestational age. Histologically, many neutrophils were present in the alveolar spaces on pulmonary sections and on stomach sections, which was in favor of a fetal acute inflammation. There was a vaginal bacteriological test positive for streptococcus of the B group during labor, which justified an antibiotic treatment. The second observation revealed a fetus of 28 WG normotroph for gestational age, with no other gestational complication. In the immediate neonatal time, the newborn developed a severe disseminated intravascular coagulation, leading to a fatal outcome at day 8. The stillborn fetus in case 3, delivered at 35 WG, had a severe growth restriction, oligohydramnios and presented a double nuchal cord.

### 3.2. Lesions Association among Placentas before the Emergence of SARS-CoV-2 Virus

We searched for the association of (1) massive and diffuse intraparenchymal dense and white-greyish fibrin deposition, (2) intervillositis and (3) trophoblastic necrosis in a case series of 8857 placentas analyzed in our institution before the emergence of SARS-CoV-2 virus. Placentas with inflammatory lesions accounted for 1240/8857 (14%) placentas analyzed in our department from January 2002 to June 2012, including chorioamniotis in 943/1240 cases (76%, 10.6% of the 8857 placentas, chronic villitis in 87/1240 cases (7%, 0.9% of the 8857 placentas) and chronic histiocytic intervillositis in 57/1240 cases (4.6%, 0.6% of the 8857 placentas).

None of these cases had a histological association composed of intraparenchymal perivillous fibrin deposition, trophoblastic necrosis and inflammatory lesions including intervillositis. Accordingly, the cumulative incidence rate of the “placentitis triad” during this period was 0% (95% CI 0–0.04%).

## 4. Discussion

Among the studies decribing histological examination of placentas in the context of SARS-CoV-2 maternal infection (on Pubmed, with keywords “SARS-CoV-2” + “placenta” + “pathology”), which include three meta-analyses, poor fetal outcome has been frequently reported in association with the “placentitis triad”, consisting in intervillositis (with an acute or chronic infiltrate), excess fibrin deposition and trophoblastic necrosis [Appendix A], refs. [3,4,5,6,8,9,10,12,13,14,15,16,17,20,21,22,25,26,27,28,29,30,31,32,33,34,35,36,37,38,39,40,41,42,43,44,45,46].

In the literature reporting this association, the intervillositis is sometimes described as sub-acute, sometimes as chronic, which was not described classically before the COVID-19 pandemic: this point could be linked to the delay between delivery and maternal SARS-CoV-2 infection. A short delay between the SARS-CoV-2 infection and the delivery, with a massive placentitis, could be linked to an acute and severe inflammation. Progressively, the neutrophils could disappear in the case of a long delay between the SARS-CoV-2 infection and the delivery. Intervillositis is a severe pathology, known to be associated with pregnancy complications, such as premature birth, fetal growth retardation and even fetal death [9]. In our three cases, it is worth noticing that the intervillositis was characterized by the presence of neutrophils in the maternal chamber infiltrate, mixed with histiocytes and lymphocytes; this pattern could be defined as subacute intervillositis. Like other reports, such as the one by Bouachba et al., our three cases were associated with recent maternal SARS-CoV-2 infection, evolutive for less than a month [10].

We can hypothesize some elements to explain variations in clinical presentations and fetal outcomes in SARS-CoV-2 infected pregnant women. In addition to the temporal variations of these lesions, a high percentage of destroyed placental tissue can lead to a more severe clinical presentation and to a poorer fetal outcome. Moreover, according to the clinical severity of the different variants of SARS-CoV-2, some differences in the obstetrical outcome, neonatal outcome and placental findings could be suspected, but are still not identified in the literature, to the best of our knowledge. We had the information of the SARS-CoV-2 variant only for the second case described, with severe maternal and fetal complications: it was the Delta variant.

In our case series of 8857 placentas, none of them had a histological association of intraparenchymal perivillous fibrin deposition, trophoblastic necrosis and inflammatory lesions including intervillositis. The three elements of the placentitis triad have been described separately in cases of stillbirth for many reasons, but never all three on the same case [47]. This suggests that the “placentitis triad”, from its lowest level, to its massive acute presentation, could be specific for SARS-CoV-2 infection, as it was not observed before the emergence of this virus in placental inflammatory pathology, and as there was no clinical or histological argument in favor of an other infectious disease, such as toxoplasmosis, rubella, cytomegalovirus infection, herpes simplex virus, syphilis or HIV. Beyond reporting three additional cases of SARS-CoV-2 maternal infection with poor fetal outcome, associated with acute and massive lesions of the “placentitis triad”, we suggest that this association of lesions could be a new placental histological feature strongly related to SARS-CoV-2 infection. Our results are in favor of a specific placental pathological signature in the context of SARS-CoV-2 infection. Nevertheless, although it also merits further work to date, we anticipate that, despite its specificity, this association would not be sensitive for the diagnosis of SARS-CoV-2 infection, but could be a placental signature of this infection in case of severe fetal outcome. Indeed, previous works have reported series of placentas in the context of SARS-CoV-2 infection with other lesions or no lesion at all.

In addition to its diagnostic specificity, the prognostic significance on pregnancy outcome of this histological association would also deserve additional works: it is possibly related to its abundance, and acute or chronic onset before delivery. Finally, deciphering the pathophysiology of these placental lesions arising in the context of SARS-CoV-2 maternal infection would also warrant further studies, as it could lead to a better management of SARS-CoV-2 infected pregnant women and their fetuses and newborns.

## 5. Conclusions

SARS-CoV-2 maternal infection could cause various histological placental lesions throughout pregnancy. The particular “placentitis triad” associating perivillous fibrin deposition, intervillositis and trophoblastic necrosis seems specific for SARS-CoV-2 infection and its acute or subacute diffuse presentation could be correlated with poor in utero but also post-natal outcome for the fetus and newborn. This finding must be confirmed by further studies.

## Figures and Tables

**Figure 1 life-13-00479-f001:**
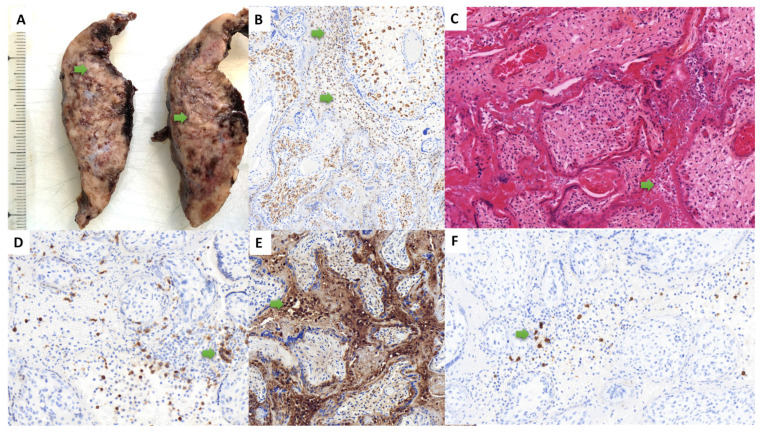
Macroscopic, histological and immunohistochemical findings of the observation #1 (at 16 weeks of gestation). Macroscopically, the parenchyma was heterogeneous, with massive fibrin deposition (**A**). Histologically, we found a trophoblast necrosis and an intervillositis (green arrow) (**C**), with an inflammatory infiltrate in the intervillous space composed mainly of neutrophils (**E**) and macrophages (**B**), and some CD3+ T Cells (**D**) and CD20+ B cells (**F**). ((**A**): macroscopic view; (**B**): anti-CD68 immunohistochemistry, ×20; (**C**): hematoxylin-eosin-saffron, ×20, (**D**): anti-CD3 immunohistochemistry, ×20; (**E**): anti-myeloperoxydase immunohistochemistry, ×20, (**F**): anti-CD20 immunohistochemistry ×20).

**Figure 2 life-13-00479-f002:**
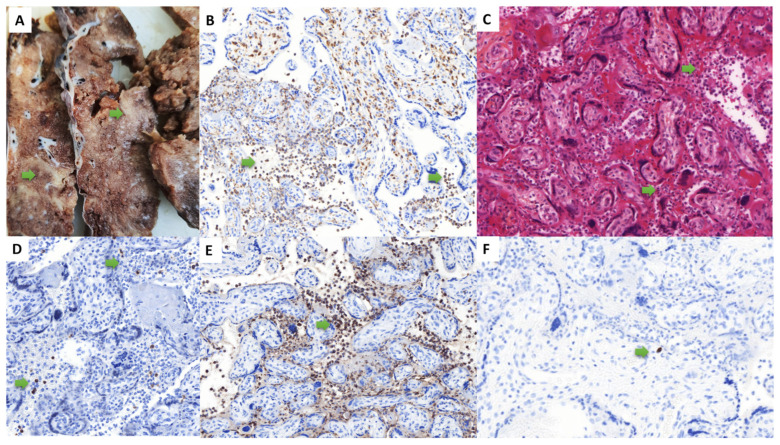
Macroscopic, histological and immunohistochemical findings of the observation #2 (at 28 weeks of gestation). Macroscopically, the parenchyma was heterogeneous, with fibrin deposition (**A**). Histologically, we found a trophoblast necrosis and an intervillositis (green arrow) (**C**), with an inflammatory infiltrate in the intervillous space composed mainly of neutrophils (**E**) and macrophages (**B**), and some CD3+ T cells (**D**), with only one CD20 positive cell, insignificant (**F**). ((**A**): macroscopic view; (**B**): anti-CD68 immunohistochemistry, ×20; (**C**): hematoxylin-eosin-saffron, ×20, (**D**): anti-CD3 immunohistochemistry, ×20; (**E**): anti-myeloperoxydase immunohistochemistry, ×20, (**F**): anti-CD20 immunohistochemistry ×20).

**Figure 3 life-13-00479-f003:**
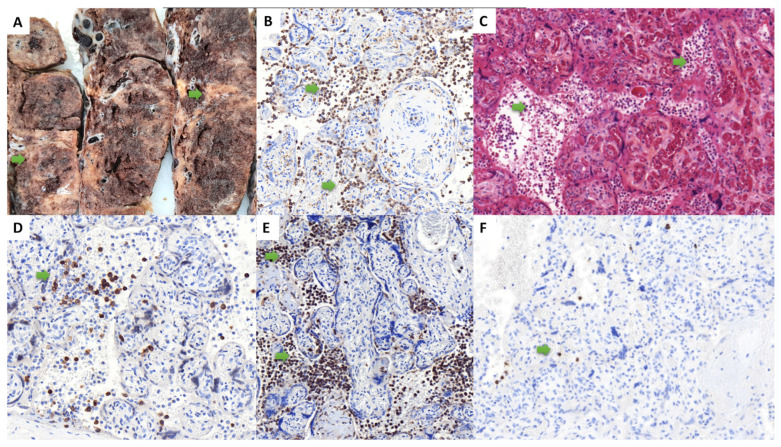
Macroscopic, histological and immunohistochemical findings of the observation #3 (at 35 weeks of gestation). Macroscopically, the parenchyma was heterogeneous, with massive fibrin deposition (**A**). Histologically, we found a trophoblast necrosis and an intervillositis (green arrow) (**C**), with an inflammatory infiltrate in the intervillous space composed mainly of neutrophils (**E**) and macrophages (**B**), some CD3+ T cells (**D**) and rare CD20+ B cells (**F**). ((**A**): macroscopic view; (**B**): anti-CD68 immunohistochemistry, ×20; (**C**): hematoxylin-eosin-saffron, ×20, (**D**): anti-CD3 immunohistochemistry, ×20; (**E**): anti-myeloperoxydase immunohistochemistry, ×20, (**F**): anti-CD20 immunohistochemistry ×20).

**Figure 4 life-13-00479-f004:**
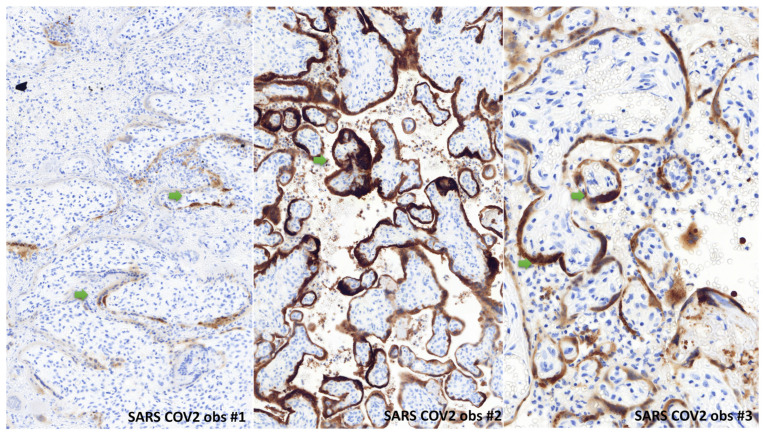
Immunohistochemical findings of the three observations with SARS-CoV-2 antibody. We observed a strong positivity (green arrow) in the three cases of the syncytiotrophoblast cytoplasm, only in the territory of placentitis triad. We can note the intervillositis in this territory. (anti-SARS-CoV-2 immunohistochemistry, ×20).

**Table 1 life-13-00479-t001:** Summary of the three cases with placental examination in the context of SARS-CoV-2 infection and poor fetal outcome. The quantitative evaluation of each antibody was classified into three categories: «low» for a minor positive staining, «moderate» for an intermediate positive staining and «high» for an abundant positive staining, or negative if no significant staining was observed.

Cases	Gestational Age (Weeks)	Days until SARS-CoV-2 Diagnosis	Clinical Presentation and Fetus Weight	Placental Weight	Massive and Diffuse Intraparenchymal Pervillous Fibrin Deposition	Intervillositis				Other Lesions
						Neutrophils (MPO+)	T Cells (CD3+)	Macrophages (CD68+)	B Cells (CD20+)	
#1	16	19	Abortion, 84 g (normal)	60 g (normal)	high	high	low	low	Low	Trophoblast necrosis No fetal lesion
#2	28	1	Maternal COVID-19 symptoms, prematurity, 1400 g (normal), neonatal distress and death at 8 days	300 g (85th percentile)	High	low	Low	high	negative	Trophoblast necrosis
#3	35	10	Cesarean for fetal distress with nuchal cord, growth retardation, 2310 g and oligoamnios	403 g (50th percentile)	high	High	high	High		Trophoblast necrosis ans a 1cm infarction and a subchorial thrombosis

## Data Availability

The data presented in this study are available on request from the corresponding author.

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
