# Peer review of "The Histopathological “Placentitis Triad” Is Specific for SARS-CoV-2 Infection, and Its Acute Presentation Can Be Associated with Poor Fetal Outcome"

_life, 2023, doi:10.3390/life13020479_

Round 1

Reviewer 1 Report

Dear Authors,

The subject addressed in the article is interesting from the perspective of the Sars-Cov-2 pandemic. The organic or functional changes of the infection at the level of internal organs allow us to evaluate the appropriate therapeutic measures to control this viral infection.

The Introduction chapter is well done and in accordance with the subject of the article.

I think that the Material and Method Chapter has big problems. First of all, it is not clear when this study was carried out. It is mentioned by the authors that the period would be: "between January 2020 and August 2021, we collected the cases associated..." but in the Results Chapter data from another period are presented: placentas analyzed in our department from January 2002 to June 2012 ". If these data want to be the control group, then it is mandatory to present the study design as well as the inclusion or exclusion criteria.

The antibodies used for IHC are listed, but the results show images from the IHC reaction for SARS-Cov-2 antibody (Fig. 4). This marker is not presented in the Method chapter.

The IHC process is briefly presented despite that the results and conclusions of the study are based on this technique.

Table no. 1 shows some IHC evaluations using a system with +/++/+++ without presenting what it represents and what the inclusion criteria are.

In the results, the authors mention: "Delta variant, which is the SARS-CoV-2 variant biologically confirmed in the second case". What are the other two virus variants of cases 1 and 3?

Data are presented in the subsection "Lesions association among placentas before the emergence of SARS-CoV-2 virus" from: "8,857 placentas analyzed" without performing a statistical analysis but only a percentage one. The data presented in this control group must have a statistical value, not a percentage one.

From this point of view, the Conclusions can present BIAS.

Reviewer 2 Report

Dear authors,

your paper is interesting, accurate and well written

the presentation of cases nice and complete

i have minor revisions to suggest

a) you've presented really nice images about histology I would like to ask you to better describe and highlight within the same images the areas where the reader must focus to identify the landmarks for the placentitis triad maybe with some circle aiming to let a reader (non expert in pathological examination) understand what to look at

2) I would like you to mention within the discussion that the three element of the placentas triad have been described in case of stillbirth for many reason (read and cite 10.36129/jog.2022.20) but of course no one of such cases due to a specific reason presented with all three at the same time as it has been documented in your  COVID cases

otherwise great job

Round 2

Reviewer 1 Report

Minor spell check required